# Theta and Gamma Activity Differences in Obsessive–Compulsive Disorder and Panic Disorder: Insights from Resting-State EEG with eLORETA

**DOI:** 10.3390/brainsci13101440

**Published:** 2023-10-10

**Authors:** Ilju Lee, Kyoung Min Kim, Myung Ho Lim

**Affiliations:** 1Department of Psychology, Dankook University, 119 Dandar-ro, Dongnam-gu, Cheonan 31116, Republic of Korea; redpstol@gmail.com; 2Department of Psychiatry, Dankook University Hospital, Cheonan 31116, Republic of Korea; profuture@naver.com; 3Department of Psychiatry, College of Medicine, Dankook University, Cheonan 31116, Republic of Korea

**Keywords:** obsessive–compulsive disorder, panic disorder, resting state, electroencephalogram, exact low-resolution electromagnetic tomography

## Abstract

**Background**: Obsessive–compulsive disorder (OCD) and panic disorder (PD) are debilitating psychiatric conditions, yet their underlying neurobiological differences remain underexplored. This study aimed to directly compare resting-state EEGs in patients with OCD and PD, without a healthy control group, using the eLORETA method. **Methods**: We collected retrospective EEG data from 24 OCD patients and 22 PD patients who were hospitalized due to significant impairment in daily life functions. eLORETA was used to analyze the EEG data. **Results**: Heightened theta activity was observed in the anterior cingulate cortex (ACC) of OCD patients compared to PD patients (PD vs. OCD, t = −2.168, *p* < 0.05). Conversely, higher gamma activity was found in the medial frontal gyrus (MFG) and paracentral lobule (PCL) in PD patients (PD vs. OCD, t = 2.173, *p* < 0.05). **Conclusions**: Our findings highlight neurobiological differences between OCD and PD patients. Specifically, the increased theta activity in the ACC for OCD patients and elevated gamma activity in the MFG and PCL for PD patients offer preliminary insights into the neural mechanisms of these disorders. Further studies are essential to validate these results and delve deeper into the neural underpinnings.

## 1. Introduction

Obsessive–compulsive disorder (OCD) is defined by the enduring presence of intrusive cognitions (obsessions) and repetitive actions or mental processes (compulsions) that individuals are driven to engage in to mitigate the distress elicited by the obsessions [1]. Panic disorder (PD) is typified by the recurrent and unanticipated occurrence of panic episodes, concomitant with pronounced feelings of apprehension and somatic manifestations such as tachycardia, dyspnea, and vertigo [1]. In the *Diagnostic and Statistical Manual of Mental Disorders, Fifth Edition (DSM-5)*, OCD has been reclassified and segregated from anxiety disorders as a result of distinctions in the underlying pathophysiological mechanisms [1,2].

Previous research has revealed that disruptions of the cortico–striato–thalamic cortical (CSTC) circuit significantly contribute to the manifestation and progression of symptoms in patients with OCD [3,4]. This circuit is comprised of interrelated connections encompassing the cortex (particularly the orbitofrontal cortex, anterior cingulate cortex, and dorsolateral prefrontal cortex), the striatum (constituted by the caudate nucleus and putamen), the thalamus, and the pallidum [5,6]. Additionally, numerous resting-state EEG studies have reported aberrant activity in the frontal brain regions. For example, increased frontal theta and reduced alpha activity have been noted in patients with OCD compared to healthy controls [7,8,9,10,11]. The results of this investigation demonstrate aberrant neuronal functioning in the anterior cingulate cortex (ACC) at rest, which aligns with findings from functional magnetic resonance imaging (fMRI) studies [12,13,14]. Researchers suggested that ACC dysfunction may be associated with the observed cognitive and affective dysregulation in patients with OCD.

Neuroanatomical models of PD also suggest that the underlying neuropathological processes contributing to panic symptoms may involve a complex interplay among limbic, cortical, and subcortical brain regions [15,16]. Previous studies revealed that notable dysfunction has been observed in both the frontal region of the brain and the hypothalamic–pituitary–adrenal (HPA) axis in patients with PD [17]. Despite the scarcity of resting-state EEG studies, a consistent pattern among PD patients has been observed, characterized by decreased alpha band activity in the frontal and parietal regions and increased beta band activity in the frontal region [18,19,20,21]. The HPA axis is a neuroendocrine system that regulates the body’s stress response, and it may disrupt the balance between excitatory and inhibitory neural activity, leading to changes in cortical activity [18,22].

As summarized above, the body of evidence investigating the neurobiological characteristics of OCD or PD continues to grow. Additionally, it would be necessary to conduct direct comparisons of neurobiological characteristics between OCD and PD patients, as any observed differences or similarities between these diagnoses could provide supplementary evidence to aid in the differential diagnosis of complex cases where the DSM-5 criteria fail to be sufficient [1]. To the best of our knowledge, no study has been identified that directly compares the neurobiological features of OCD and PD.

Recently, great interest has been given to spontaneous brain activity during the task-negative experiment (i.e., resting state) in patients with neuropsychiatric disorders [23,24]. EEG recordings, obtained while patients are at rest, have been cited as valuable diagnostic tools for various psychiatric disorders, including neurodevelopmental disorders [24].

Various methods have been employed to analyze resting-state EEG data in assessing neuropsychiatric disorders. Exact low-resolution electromagnetic tomography (eLORETA) is a three-dimensional, discrete, linear, and weighted minimal norm inverse solution technique [25]. This method is characterized by its capacity to precisely localize an EEG source in brain regions, albeit with low spatial resolution. Due to its compliance with the principles of linearity and superposition, eLORETA produces a low-resolution approximation of any distribution of electrical neuronal activity. An extensive comparison with other linear inverse methods revealed that eLORETA exhibits improved localization capabilities in the presence of noise and situations with multiple sources [25].

The aim of this study is to compare resting-state EEG activity using the eLORETA method between patients with OCD and PD in order to investigate potential differences in EEG patterns between the two disorders. We used a retrospective design to collect resting-state EEG data from OCD and PD patients who were hospitalized for the first time in their lives due to significant impairment in daily life functions. Before directly comparing the resting-state EEG of OCD and PD patients, we first sought to characterize the EEG features of our participants using a normative database adjusted for age and gender. Subsequently, we conducted a whole-brain voxel-by-voxel analysis to compare the resting-state EEG patterns between the two patient groups. Based on previous research, our hypothesis posits that there will be increased theta band activity in the frontal region of OCD patients and heightened upper beta band activity in PD patients.

## 2. Materials and Methods

We utilized a database from Dankook University Hospital, Republic of Korea, which stored retrospective EEG data of inpatients admitted from 2010 to 2017 as part of their admission protocol for epilepsy evaluation. The EEG tests were performed during the daytime, and of the total 40 min of testing time, only 6 min of the resting state with closed eyes was extracted and stored. All patients were diagnosed by a board-certified psychiatrist using the diagnostic criteria from the DSM-5 and the International Statistical Classification of Diseases and Related Health Problems (ICD-10) section addressing mental, behavioral, and neurodevelopmental disorders. Initially, we recruited 53 OCD patients and 57 PD patients. The exclusion criteria for our study were as follows: (1) patients who were not admitted for the first time, (2) those who were referred for a neurology consultation during their admission, and (3) patients with additional diagnostic codes registered other than OCD and PD. From the total dataset, 31 patients with more than one hospitalization experience were excluded (20 OCD patients and 11 PD patients). Additionally, 3 PD patients who were referred for neurology consultations, 25 patients with additional diagnostic codes suggesting comorbid conditions (9 OCD patients and 14 PD patients), and 7 PD patients for whom resting-state EEG analysis was not feasible were also excluded. Consequently, we collected data from 24 OCD (ICD-10 code F42; mean age ± standard deviation (SD): 28.20 ± 6.82) patients and 22 PD (ICD-10 code F410; mean age ± SD: 30.54 ± 7.90) patients for our study. The gender distribution for the OCD group was 11 males and 13 females, while the PD group consisted of 10 males and 12 females. All patients underwent the EEG protocol prior to hospitalization. This retrospective study was approved by the ethical review committee of Dankook University Hospital (IRB No. 2018-02-008).

### 2.1. EEG Data Collection and Preprocessing

The experiment was conducted in a sound-attenuated room. Resting-state EEG data were recorded in two 3 min intervals with eyes closed (i.e., EC1-EC2, totaling 6 min). The Comet-PLUS XL EEG system (Natus, Middleton, WI, USA) was employed, with left and right earlobes serving as reference channels for quantitative EEG. Raw EEG signals were acquired from 19 scalp positions (Fp1, Fp2, F7, F3, Fz, F4, F8, T3, C3, Cz, C4, T4, T5, P3, Pz, P4, T6, O1, and O2) following the International 10–20 System. Experts placed electrodes on each participant to ensure accurate channel placement. Data were sampled at a rate of 400 Hz and filtered within a 0.5 to 100 Hz range. Impedances for all channels were kept below 10 kΩ. Participants remained awake and seated comfortably, instructed to relax and minimize body movement. All data were exported in ASCII format using TWin software (Natus, Middleton, WI, USA). EEG preprocessing was performed with MATLAB R2015a (MathWorks, Natick, MA, USA) and EEGLAB v14.0. In the offline analysis, the EEG data were filtered using a 1–50 Hz bandpass filter for the analysis of interest. Visual inspection was employed for offline artifact rejection to remove epochs attributable to body movement or poor channel quality. Ocular artifacts, such as eye blinks and eye rolls, were removed based on the ocular channel. Finally, independent component analysis was performed to eliminate ocular and significant muscle artifacts [26].

### 2.2. Utilization of Normal Database

Our clinical investigation analyzed the EEG patterns of patients diagnosed with OCD and PD by contrasting their data with the Lifespan Normative Database (Applied Neuroscience, Inc., St. Petersburg, FL, USA). This esteemed database contains EEG recordings, taken during both eyes-open and eyes-closed states, from healthy normal individuals ranging in age from birth to 82 years [27]. Our emphasis was notably on the Z-score statistics, grounded in the standard normal distribution with a mean of zero and a variance of one. These Z-scores delineate the degree to which an individual’s EEG deviates from the database's normative values set for age and gender-matched peers.

### 2.3. EEG Source Localization and Statistical Analysis

eLORETA was used to compute the intracortical distribution of the electrical activity from the surface EEG data [25]. This method is a discrete, three-dimensionally distributed, linear weighted minimum norm inverse solution. The weights used in eLORETA provide tomography with the property of exact localization to test point sources, yielding images of current density with exact localization, albeit with low spatial resolution (i.e., neighboring neuronal sources are highly correlated). A further property of eLORETA is that it has no localization bias, even in the presence of structured noise [28].

This study restricted the solution space to the cortical gray matter, corresponding to 6239 voxels at a spatial resolution of 5 × 5 × 5 mm. The Montreal Neurological Institute average MRI brain (MNI152) was used as a realistic head model, for which the lead field was computed [29,30]. The validity of eLORETA tomography was confirmed in previous validation studies of LORETA and sLORETA [31,32].

Selected artifact-free EEG segments were used to calculate the eLORETA intracranial spectral density from 1 to 50 Hz with a resolution of 1 Hz. Functional eLORETA images of spectral density were computed for five frequency bands: delta (1–4 Hz), theta (4.5–8 Hz), alpha (8.5–12 Hz), beta (13–30 Hz), and gamma (30.5–40 Hz).

The difference in cortical oscillations in each frequency band and the localized source between groups was assessed voxel-by-voxel using the independent samples t-test and the paired t-test based on eLORETA log-transformed current density power. In the resulting statistical three-dimensional images, cortical voxels showing significant differences were identified using a non-parametric approach (statistical non-parametric mapping, SnPM) via randomizations [33]. This randomization strategy determined the critical probability threshold values for the observed t-values with correction for multiple comparisons across all voxels and all frequencies. By evaluating the empirical probability distribution of the “maximal statistics” in the null hypothesis, permutation and randomization tests have been demonstrated to be effective in controlling Type I errors in neuroimaging studies [33]. Demographic data (gender and age) were examined using the SPSS version 26 (IBM Corp., Armonk, NY, USA).

## 3. Results

### 3.1. Demographic Features

Table 1 provides demographic information for the patients, including gender and age. The groups were matched for gender and age, with 11 males in the OCD group and 10 males in the PD group, and average ages of 28.20 ± 6.82 and 30.54 ± 6.90, respectively.

### 3.2. Comparison with Normative EEG Database

In this exploratory study, the resting-state EEG of each participant was compared with a normative database to compute the corresponding Z-scores. Figure 1 provides a visual representation of these comparisons, displaying the EEG channels across various frequencies—delta, theta, alpha, beta, and high beta—against the normative database.

Our preliminary findings hint at potential trends in the EEG frequencies associated with the two disorders. For the OCD patient group, there seems to be a subtle trend of elevated theta Z-scores in the frontal EEG channels (absolute theta *Z*-Score: F7 = 1.647; F8 = 1.556; FP1 = 1.365; F4 = 1.325). Similarly, the PD patient group exhibited a slight trend of increased high beta Z-scores in the fronto-central regions (absolute high beta *Z*-Score: Fz = 1.347; Cz = 1.204; P3 = 1.180; Pz = 1.116). A more detailed account of these observations can be found in Appendix A.

### 3.3. Comparison of Source Localization between OCD and PD Patients in the Resting State

Resting-state EEG neuroimaging findings revealed significant differences between PD and OCD patients using whole-brain voxel-by-voxel unpaired t-tests based on a SnPM generated via eLORETA. After 5000 randomizations and permutations, significant t-values were observed (PD vs. OCD; max *t* threshold ±2.14, *p* < 0.05) and represented on a color scale and Brodmann area (BA) in Figure 2 and Figure 3. The MRI slices are located according to MNI space coordinates, which correspond to the most significant voxel.

Figure 2 shows the statistical differences in current source density (CSD) between the OCD and PD patients in resting-state eye-closed conditions, representing the differences in theta activity. Source localization analysis showed significantly higher CSD values for theta band activity in the ACC (BA 24; *t* = −2.168, *p* < 0.05; BA 32; *t* = −2.151, *p* < 0.05) in the OCD group compared to PD.

These EEG source analysis results also demonstrated significant differences in the PD patient group. Figure 3 represents the differences in gamma activity in the MFG (BA 6; *t* = 2.173, *p* < 0.05, corrected for multiple comparisons) and PCL (BA 4; *t* = 2.164, *p* < 0.05) of PD patients. These results are corrected for multiple comparisons using non-parametric randomization.

## 4. Discussion

The primary objective of our study was to discern potential neurobiological differences between OCD and PD patients by analyzing resting-state EEGs using the eLORETA method. We posited that OCD patients would exhibit an uptick in frontal theta activity, while PD patients would display increased upper beta activity. Our results highlighted a trend of increased theta activity in the anterior cingulate cortex (ACC) for OCD patients compared to PD and elevated gamma activity in the medial frontal gyrus (MFG) and paracentral lobule (PCL) for PD patients compared to OCD. Furthermore, a noticeable trend emerged when leveraging Z-scores for comparison with a normative control: an increase in theta activity for OCD and a surge in high beta activity for PD. It is imperative to note that our study is exploratory in nature, and the observations primarily offer preliminary insights into the neuropsychological underpinnings of both OCD and PD. The resting-state EEG patterns might reflect the neurobiological characteristics of OCD and PD.

### 4.1. Increased Theta Band Activity in the Frontal Region of OCD Patients

In our study, the increased theta band activity observed in the ACC among OCD patients, even when compared to PD patients, aligns with previous studies reporting similar findings. The role of ACC in cognitive and affective dysregulation in OCD is well established, making it a crucial component of the CSTC circuit implicated in the pathophysiology of OCD [3,4,14]. The ACC is involved in various cognitive processes, such as error detection, conflict monitoring, and decision-making, often impaired in OCD patients [13,14,34,35]. Hyperactivity in the ACC may produce persistent high error or conflict signals that are hypothesized to drive the compulsive behaviors characteristic of OCD [34,35,36]. This heightened error sensitivity and persistent engagement in compulsions contribute to the chronicity and severity of OCD symptoms.

Additionally, the role of other structures, especially the subthalamic nucleus (STN), should be emphasized. The STN, an integral component of the CSTC circuit, has been shown to have oscillatory coupling with the ACC, and this coupling is particularly significant in understanding the intricate neural dynamics in OCD. The oscillatory activity of the STN, especially in the theta band, has been associated with the severity of OCD symptoms. Specifically, increased theta activity in the STN was found in patients with severe OCD. Following the deep brain stimulation (DBS) intervention, there was a noticeable reduction in theta oscillations in the STN, which correlated with a decrease in the severity of OCD symptoms [37,38]. This suggests that the STN’s theta activity may play a pivotal role in the manifestation of OCD symptoms and that modulating this activity can have therapeutic implications.

Increased theta activity in the ACC may suggest a hypofunction in this brain region, contributing to the observed symptoms. Several EEG studies investigating OCD’s pathophysiology have reported significant low-frequency band (delta and theta) findings in the ACC compared to healthy controls [11,39,40,41,42,43,44]. Cavanagh et al. examined the link between OCD symptoms and brain activity during action-monitoring tasks, including resting-state and error-related tasks [42]. Their study revealed that OCD symptoms correlate with functional differences in medio-frontal systems, specifically, increased activity in the rostral anterior cingulate cortex (rACC) at rest. Additionally, they found that interactive ACC systems associated with avoiding maladaptive actions remain intact in the high OCD symptom group. These results are corroborated by a recent extensive investigation conducted on early-onset OCD in youth [43]. Kamaradova and colleagues compared resting-state EEG data from 20 OCD patients and 15 healthy controls, analyzing cortical EEG sources across various frequency bands [44]. Their results revealed higher frontal delta and theta EEG activity in OCD patients, especially those with cognitive impairment. Additionally, elevated theta band activity has been linked to resistance to selective serotonin reuptake inhibitors (SSRIs) in OCD patients [8,9]. Hence, elevated frontal theta activity may be regarded as an EEG feature indicative of central nervous system vulnerability in patients with OCD [11].

### 4.2. Increased Gamma Band Activity in the Medial Frontal Gyrus and Paracentral Lobule

In the present study, higher gamma band activity was observed in the MFG and PCL regions among PD patients compared to the OCD group. Although the results of this study do not directly support the previous findings of increased beta band activity in the frontal region of PD patients, the results are deemed reflective of the pathophysiological features of PD.

Hyperactivation of the amygdala may offer a plausible explanation for our findings. The amygdala is a key brain region involved in processing emotions, particularly fear and anxiety [45]. It plays an essential role in detecting emotionally salient stimuli and generating emotional responses [46]. Hyperactivation of the amygdala has been observed in various anxiety disorders, including PD, generalized anxiety disorder (GAD), and post-traumatic stress disorder [47].

Cortical gamma oscillations are fast neuronal oscillations (typically around 30–100 Hz) that have been associated with various cognitive processes, including attention, perception, learning, and memory [48]. These oscillations are thought to play a crucial role in integrating and synchronizing information across different brain regions during cognitive processing [49]. Impaired sensory gating in PD patients might be related to the increased gamma activity in the MFG and PCL, leading to difficulties in filtering out irrelevant sensory information and heightened sensitivity to sensory input [50]. This impairment in sensory gating could contribute to the heightened stress response and anxiety experienced by PD patients as they struggle to effectively process and manage incoming sensory information [51,52]. Consequently, the increased gamma activity in the MFG and PCL regions might serve as a neural signature of the dysregulated sensory processing mechanisms in PD patients, potentially providing valuable insights into the underlying neural mechanisms of panic disorder and guiding future research and therapeutic interventions [53]. For instance, Miskovic and Keil reviewed electrophysiological studies on human classical conditioning and found that fear-conditioned stimuli increase gamma oscillatory activity and enhance sensory processing during early stimulus evaluation [52]. The authors suggested that the amygdala and frontal region mediate these fear-related changes in cortical sensory processing. Moreover, studies have shown that heightened amygdala activity in response to emotionally salient stimuli can modulate cortical gamma oscillations, facilitating the processing and integration of emotional information [51]. This modulation may occur through direct connections between the amygdala and various cortical regions, including the prefrontal cortex, insula, and sensory cortices [54]. Impairment in this sensory processing mechanism is thought to heighten sensitivity to sensory input, such as the interoceptive senses, leading to increased anxiety and panic symptoms even at rest [15,50,51].

Our study has several limitations that warrant consideration. Firstly, due to the clinical nature of our research, we utilized a 19-channel EEG setup. While LORETA source modeling has demonstrated commendable outcomes when paired with various brain imaging modalities and even with 19-channel EEGs [55,56,57], it is important to acknowledge that the precision of source localization might be less optimal compared to modern high-definition EEGs. Secondly, while our study leveraged a normative database controlled for age and gender to compare with our clinical samples, we were only able to identify certain trends. For subsequent research, it would be beneficial not only to account for age and gender but also to incorporate other potential confounders, such as intelligence and clinical variables. Multivariate investigations that consider factors that might influence symptom severity are helpful for a more comprehensive understanding. Thirdly, the sample size was relatively small, which may limit the generalizability of the findings. Larger, more diverse samples are needed to increase confidence in the results and ensure they are representative of the broader OCD and PD populations. Lastly, our study had limitations in capturing certain clinical variables of the inpatients. While our EEG database provided insights into their diagnoses and the number of hospitalizations, it was primarily intended for epilepsy evaluations. Consequently, sensitive information such as medication administration and clinical scales could not be ascertained. This lack of detailed clinical information could influence the interpretation of the EEG findings in relation to the clinical manifestations of OCD and PD.

It is essential to note that the current study’s findings are preliminary and should be interpreted with caution. Further research is needed to corroborate these results and examine the underlying neural mechanisms in more detail. Future studies might consider utilizing larger sample sizes, more diverse patient populations, and the inclusion of healthy control groups for comparison. Additionally, examining the relationship between EEG findings and clinical symptom severity or treatment response could provide valuable insights into the clinical utility of these neurobiological markers in managing OCD and PD.

## 5. Conclusions

This study adds to the growing body of literature highlighting the neurobiological differences between OCD and PD patients. The findings of increased theta activity in the ACC of OCD patients and increased gamma activity in the MFG and PCL of PD patients provide preliminary insights into the potential neural mechanisms underlying these disorders. By identifying distinct neural signatures, this research may contribute to improved diagnostic accuracy and pave the way for more targeted therapeutic interventions in the future. However, further research is needed to corroborate these findings, investigate the causal relationships between neural differences and clinical symptoms, and ultimately enhance our understanding of the neurobiological underpinnings of OCD and PD.

## Figures and Tables

**Figure 1 brainsci-13-01440-f001:**
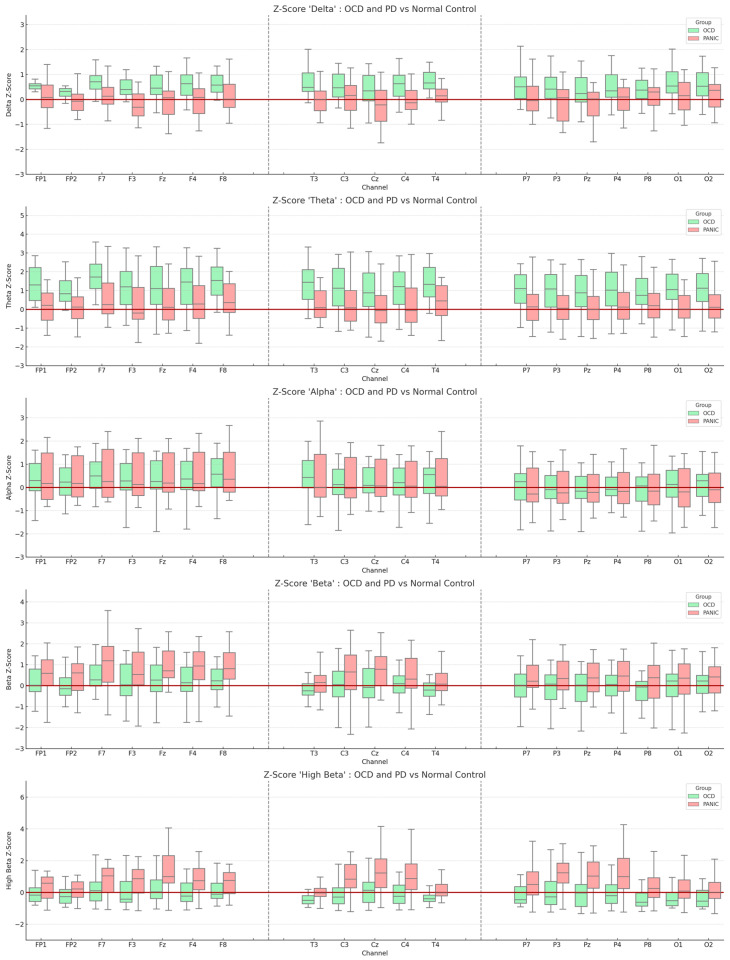
Boxplot representation of Z-scores derived from the normative database. The light green boxes indicate the OCD group, while the red boxes represent the PD group. The *y*-axis displays the average Z-scores for each frequency, and the *x*-axis represents the measured EEG channels. The red line intersecting the *y*-axis denotes a Z-score of zero. All values are means, and the error bars depict standard deviations.

**Figure 2 brainsci-13-01440-f002:**
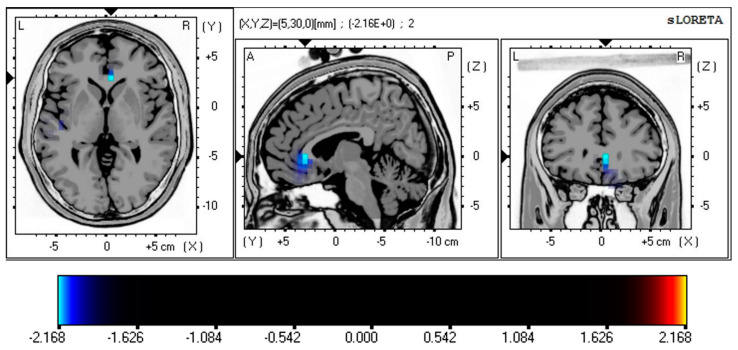
Comparison of source localization between the OCD and PD group in resting-state condition. Colored areas represent the spatial extent of voxels with a significant difference between OCD and PD groups. The *t*-values are displayed on a color scale (PD vs. OCD; max *t* threshold ±2.14, *p* < 0.05), where cyan/blue shades denote heightened neuronal activity in the theta frequency band (4–8 Hz) within the ACC (BA 24; *t* = −2.168, *p* < 0.05; BA 32; *t* = −2.151, *p* < 0.05, corrected for multiple comparisons) of OCD patients. The structural anatomy is illustrated in grayscale, with abbreviations indicating direction (A—anterior; P—posterior; L—left; R—right). Abbreviation: SnPM, statistical non-parametric map; eLORETA, exact low-resolution electromagnetic tomography; MNI, the Montreal Neurological Institute average MRI brain; ACC, anterior cingulate cortex; BA, Brodmann area; PD, panic disorder; OCD, obsessive–compulsive disorder.

**Figure 3 brainsci-13-01440-f003:**
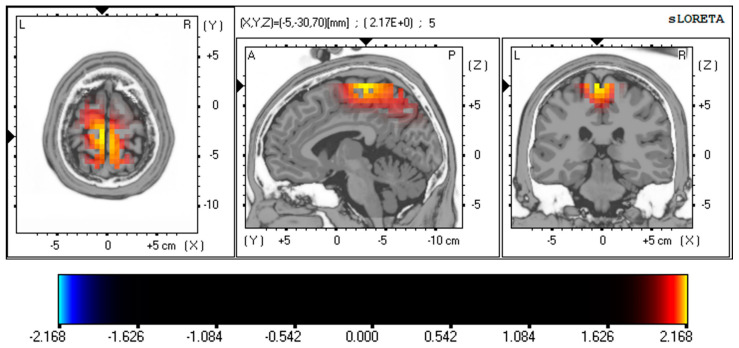
Resting-state EEG findings reveal distinctions between PD and OCD using whole-brain, voxel-wise unpaired *t*-tests, drawing on a SnPM generated through eLORETA. The images are set according to MNI space coordinates associated with the most notable voxel. The *t*-values appear on a color scale (PD vs. OCD; max *t* threshold ±2.14, *p* < 0.05), with yellow/red shades signifying increased neuronal activity in the gamma frequency band (30–40 Hz) in the MFG (BA 6; *t* = 2.173, *p* < 0.05, corrected for multiple comparisons) and PCL (BA 4; *t* = 2.164, *p* < 0.05, corrected for multiple comparisons) of PD patients. The structural anatomy is depicted in grayscale, using abbreviations for direction (A—anterior; P—posterior; L—left; R—right). Abbreviation: SnPM, statistical non-parametric map; eLORETA, exact low-resolution electromagnetic tomography; MNI, the Montreal Neurological Institute average MRI brain; MFG, medial frontal gyrus; PCL, paracentral lobule; BA, Brodmann area; PD, panic disorder; OCD, obsessive–compulsive disorder.

**Table 1 brainsci-13-01440-t001:** Demographics of OCD and PD patients.

	OCD		PD			
	(*n* = 24)		(*n* = 22)			
	M	SD	M	SD	df	*t*-Test
**Demographics**						
Age	28.20	±6.82	30.54	±7.90	45	−0.288
Gender (M/F)	11/13	-	10/12	-		-

## Data Availability

The resting-state EEG data used to support the findings of this study are available from the corresponding author upon request.

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
