# Peer review of "Theta and Gamma Activity Differences in Obsessive–Compulsive Disorder and Panic Disorder: Insights from Resting-State EEG with eLORETA"

_brainsci, 2023, doi:10.3390/brainsci13101440_

Round 1

Reviewer 1 Report

The retrospective data analysis reveals the neurobiological differences between OCD and PD patients.

The authors found an increase of theta activity in the ACC of OCD patients and an increase of gamma activity in the MFG and PCL of PD patients.

However, I cant suggest the publication of this submission due to the following reasons:

1) There was no a healthy control.

2) Source location was based on a brain template rather than individual MRI, so the result cannot be reliable, additionally, there was only 19 channels for source analysis which was not acceptable.

3) the behavioral/clinical correlate of the EEG alteration remained unknown.

Author Response

1) There was no healthy control.

Response: We appreciate your comment regarding the inclusion of a healthy control group. The primary objective of our study was to discern the potential neurobiological differences between OCD and PD patients. To address the lack of a direct healthy control group, we utilized a well-established normative EEG database that was adjusted for age and gender to compare the EEG patterns of our clinical samples. You can find it on lines 85~87, and lines 136 ~ 144. It is highlighted in yellow.

We understand the importance of a control group and will certainly consider incorporating it in future studies to offer a more comprehensive comparison.

2) Source location was based on a brain template rather than individual MRI, so the result cannot be reliable, additionally, there was only 19 channels for source analysis which was not acceptable.

Response: Thank you for pointing out these concerns. We recognize that using individual MRI data for source localization offers potential advantages over using a brain template. In this study, due to the clinical nature of our data collection, we relied on the eLORETA method and a brain template. Regarding the 19-channel EEG setup, it was employed due to the clinical context of our research. Based on the previous studies, eLORETA source modeling has demonstrated exploratory outcomes with a variety of brain imaging modalities, even with 19-channel EEGs. We added the references in the limitation section. You can find it on lines 322~327 and it is highlighted in yellow.

Here are the references that utilized a 19-channel EEG study:

  • Horacek, J.; Brunovsky, M.; Novak, T.; Skrdlantova, L.; Klirova, M.; Bubenikova-Valesova, V.; Krajca, V.; Tislerova, B.; Kopecek, M.; Spaniel, F.; et al. Effect of Low-Frequency rTMS on Electromagnetic Tomography (LORETA) and Regional Brain Metabolism (PET) in Schizophrenia Patients with Auditory Hallucinations. Neuropsychobiology 2007, 55, 132-142, doi:10.1159/000106055.
  • De Ridder, D.; Vanneste, S.; Kovacs, S.; Sunaert, S.; Dom, G. Transient alcohol craving suppression by rTMS of dorsal anterior cingulate: An fMRI and LORETA EEG study. Neurosci. Lett. 2011, 496, 5-10, doi:https://doi.org/10.1016/j.neulet.2011.03.074.
  • Lanzone, J.; Imperatori, C.; Assenza, G.; Ricci, L.; Farina, B.; Di Lazzaro, V.; Tombini, M. Power Spectral Differences between Transient Epileptic and Global Amnesia: An eLORETA Quantitative EEG Study. Brain Sci 2020, 10, doi:10.3390/brainsci10090613.

Nonetheless, we acknowledge that the precision of source localization might be enhanced with modern high-definition EEGs. We'll bear this in mind for future studies.

3) the behavioral/clinical correlate of the EEG alteration remained unknown.

Response: We value this observation. The primary focus of our study was to identify potential neurobiological differences in EEG patterns between OCD and PD patients. While we did not directly investigate the relationship between EEG alterations and clinical/behavioral manifestations, we concur that such an exploration would provide a richer understanding of the disorders. Our EEG database provided insights into the diagnosis and hospitalization frequency of the patients, but more detailed clinical data, such as medication administration and clinical scales, were not available. We understand the importance of these clinical correlates and will consider integrating them in future investigations. We have addressed this in the limitations section, which can be found on lines 335-341.

Reviewer 2 Report

In this cross sectional retrospective work the authors addressed the comparison between neurophysiological markers of resting state of two patient cohorts OCD and PD.  The paper might be potentially interesting however some critical issues remain:

-I strongly recommend inclusion of a healthy patient cohort to establish meaningful comparisons. As it is now, the design is not correct.

-Please clearly state when was the data collected? 

-Please provide years of disease since diagnosis for each patient. Were patients recorded suffer from depression,  anxiety etc?

-Were PD and OCD patients recorded under medications? please clarify and provide detailed information for each patient.

-For PD patients please provide UPDRS scores. 

-Please provide details on the number of ICA components removed from each data set.

-Why was AV reference used for analysis?

-Given the limited number of electrodes (19), I am not convinced if it really makes sense to conduct a source localization analysis. Probably a power o other analysis would be more useful in this case.  Could you please estimate your the error as you used 19 electrodes instead of a reasonable number say 64 or 128 channels. 

-Please also clearly state as limitation that a standardized head model was used instead of MRI for each participant.

This study aims to compare the neurophysiological resting state markers of two patient cohorts (OCD and PD). The study would be potentially interesting however there are critical concerns that prevent consideration of this article. 

-First, the number of electrodes recorded is not sufficient for a source localization analysis.

-A control age-matched control group is missing.

-there are patient details that are missing. Also wondering if the patient cohorts happen to be  too heterogenous and so confounding actors may biased  the results?

Author Response

  1. Inclusion of a healthy patient cohort
    Response: We appreciate your suggestion regarding the inclusion of a healthy control group. As our primary focus was to discern potential neurobiological differences between OCD and PD patients, we utilized a well-established normative EEG database adjusted for age and gender to compare with our clinical samples, as mentioned on lines 85-87 and 136-144. Also, we have addressed this in the limitations section, which can be found on lines 327-332. We understand the importance of having a direct healthy control group and will consider this in future studies for a more comprehensive comparison.

  2. Data collection timeframe
    Response: Thank you for pointing this out. This data spanning from 2010 to 2017. This information has been added to the methods section on lines 93 ~ 97.

  3. Years of disease since diagnosis and other conditions 
    Response: We acknowledge the importance of this information. Unfortunately, our EEG database primarily provided insights into the diagnosis and frequency of hospitalizations and did not include the duration since diagnosis or details on other conditions such as depression or anxiety. We have addressed this in the limitations section, which can be found on lines 335-341. We will consider capturing this information in future studies.

  4. Medication details for OCD and PD patients
    Response: We understand the potential influence of medication on EEG readings. Due to the retrospective nature of our study, the precise medication details for each patient were not available in our database. However, we are aware of the significance of this data and will ensure its inclusion in future research. We have addressed this in the limitations section on same line response #3.

  5. UPDRS scores for PD patients
    Response:
    Thank you for raising this point. Unfortunately, our current dataset does not include UPDRS scores for PD patients. We recognize the importance of such clinical scales, and we will consider their inclusion in subsequent investigations.

  6. Number of ICA components removed 
    Response: In our preprocessing steps, we took care to ensure that the number of ICA components removed did not compromise the integrity of the data, given the use of a 19-channel EEG. We excluded participants from the study if more than three ICA components had to be removed. This exclusion was predominantly observed in the PD patient group. On average, two ICA components, primarily associated with frontal muscle movement or eye rolling, were removed per participant. For specific details regarding our exclusion criteria, please refer to the method section on lines 100 to 110. The detailed breakdown of ICA components removed for each participant can be found in Supplementary 2.

  7. Use of AV reference
    Response:
    We apologize for the oversight. In our study, we actually used linked ear references, not an average reference. The error in the manuscript was a typographical mistake, and we have since corrected it. You can verify this change on lines 129 to 130. Thank you for pointing it out.

    8 & 9. Concerns about the number of electrodes for source localization and Use of a standardized head model
    Response: Thank you for pointing out these concerns. We recognize that using individual MRI data for source localization offers potential advantages over using a brain template. In this study, due to the clinical nature of our data collection, we relied on the eLORETA method and a brain template. Regarding the 19-channel EEG setup, it was employed due to the clinical context of our research. Based on the previous studies, eLORETA source modeling has demonstrated exploratory outcomes with a variety of brain imaging modalities, even with 19-channel EEGs. We added the references in the limitation section. You can find it on lines 322~327 and it is highlighted in yellow.

    Here are the references that utilized a 19-channel EEG study:

    • Horacek, J.; Brunovsky, M.; Novak, T.; Skrdlantova, L.; Klirova, M.; Bubenikova-Valesova, V.; Krajca, V.; Tislerova, B.; Kopecek, M.; Spaniel, F.; et al. Effect of Low-Frequency rTMS on Electromagnetic Tomography (LORETA) and Regional Brain Metabolism (PET) in Schizophrenia Patients with Auditory Hallucinations. Neuropsychobiology 2007, 55, 132-142, doi:10.1159/000106055.
    • De Ridder, D.; Vanneste, S.; Kovacs, S.; Sunaert, S.; Dom, G. Transient alcohol craving suppression by rTMS of dorsal anterior cingulate: An fMRI and LORETA EEG study. Neurosci. Lett. 2011, 496, 5-10, doi:https://doi.org/10.1016/j.neulet.2011.03.074.
    • Lanzone, J.; Imperatori, C.; Assenza, G.; Ricci, L.; Farina, B.; Di Lazzaro, V.; Tombini, M. Power Spectral Differences between Transient Epileptic and Global Amnesia: An eLORETA Quantitative EEG Study. Brain Sci 2020, 10, doi:10.3390/brainsci10090613.

    Nonetheless, we acknowledge that the precision of source localization might be enhanced with modern high-definition EEGs. We'll bear this in mind for future studies.

Reviewer 3 Report

This is an interesting comparative EEG study using LORETA between patients with OCD and PD well-written and analyzed.

I have some questions about the inclusion criteria, the patients were taking psychiatric medications and all the patients had normal neuroimaging studies. CT or MRI or both?

New onset diagnoses or old patients? 

Author Response

Inclusion Criteria & Medications
Response:
The inclusion criteria for this study primarily involved patients diagnosed with either OCD or PD who were hospitalized for the first time in their lives due to significant impairment in daily life functions. As our EEG database was primarily intended for epilepsy evaluations, detailed information regarding medication administration for psychiatric conditions was not accessible. We have addressed this in the Materials and Methods section, which can be found on lines 100-111. We recognize the potential influence of medication on EEG findings and will consider incorporating such data in future studies.

New onset diagnoses or old patients? 
Response: The patients included in our study were those hospitalized for the first time due to their respective disorders (OCD or PD). This was an intentional criterion to reduce potential variability introduced by long-term treatments or the progression of the disorder over an extended period. We aimed to capture a snapshot of the neurobiological underpinnings of these disorders closer to their onset, albeit recognizing that the exact time of onset might precede the first hospitalization.

Reviewer 4 Report

Hello Dears;

Thanks for the research.

Comments:

1- The number of samples is not enough for this conclusion

2-A valid questionnaire was not used for diagnosis and only the doctor's diagnosis was used, which may be wrong.

Author Response

Thank you for your valuable feedback. We have carefully considered each of your comments.

1- The number of samples is not enough for this conclusion
Response:
We acknowledge the limitation regarding the sample size in our study. Our primary aim was to offer a preliminary exploration into the potential neurobiological differences between OCD and PD patients. While the trends we observed are interesting and align with some prior research, we emphasize that our study is exploratory in nature. In our discussion and limitation sections (lines 322-349), we have highlighted the need for larger, more diverse samples to increase confidence in the results and ensure they are representative of the broader OCD and PD populations.

2- A valid questionnaire was not used for diagnosis and only the doctor's diagnosis was used, which may be wrong
Response:
 We value this observation. The primary focus of our study was to identify potential neurobiological differences in EEG patterns between OCD and PD patients. While we did not directly investigate the relationship between EEG alterations and clinical/behavioral manifestations, we concur that such an exploration would provide a richer understanding of the disorders. Our EEG database provided insights into the diagnosis and hospitalization frequency of the patients, but more detailed clinical data, such as medication administration and clinical scales, were not available. We understand the importance of these clinical correlates and will consider integrating them in future investigations. We have addressed this in the limitations section, which can be found on lines 335-341.

Thank you for drawing our attention to these issues. We believe that addressing these concerns will enhance the quality and robustness of our research.

Round 2

Reviewer 1 Report

After adding the healthy EEG data derived from a public dataset, the paper is much improved. The article can be accepted in the current form.

Author Response

After adding the healthy EEG data derived from a public dataset, the paper is much improved. The article can be accepted in the current form.

Response: Thank you for recognizing the improvements in our manuscript. We appreciate your positive feedback and are pleased that the added EEG data enhanced the paper's quality.

Best regards,

Ilju, Lee. 

Reviewer 2 Report

Thanks so much for considering my concerns. While the authors acknowledge the absence of a control group as limitation of their study, it is still relevant to make it clear in the abstract by indicating that this is a comparative study between two disease cohorts with no healthy control group.  

with regards to the OCD group, the authors emphasized an interesting finding concerning the ACC. However other structures such as the STN have been implicated in OCD (Please see https://doi.org/10.1093/brain/awx164)

Please discuss about this.

The language quality of the manuscript is appropriate. 

Author Response

Clarify Absence of Control Group in Abstract

Reponse: Thank you for your insightful comments and for highlighting the importance of clarifying the study design in the abstract.

In response to your feedback, we have revised the abstract to emphasize that this research is a direct comparison between two disease cohorts, namely obsessive-compulsive disorder (OCD) and panic disorder (PD), and that it does not include a healthy control group. Specifically, the updated abstract now states: "This study aimed to directly compare resting-state EEGs in patients with OCD and PD, without a healthy control group, using the eLORETA method."

Addressing the Implication of STN in OCD Pathophysiology

Response: Thank you for your insightful comment pointing out the importance of not only emphasizing the ACC but also considering other pivotal structures such as the STN in the context of OCD.

In response to your feedback, we have revisited and revised Section 4.1 of our discussion(4.1. Increased theta band activity in the frontal region of OCD patients) to provide a more comprehensive understanding of the role both the ACC and the STN play in OCD. The revised content can be found on lines 259 to 279, highlighted in yellow for your convenience. We have additional references which provide the oscillatory coupling of the STN and its implication in OCD:

  1. Wojtecki, L.; Hirschmann, J.; Elben, S.; Boschheidgen, M.; Trenado, C.; Vesper, J.; Schnitzler, A. Oscillatory coupling of the subthalamic nucleus in obsessive compulsive disorder. Brain 2017, 140, e56-e56, doi:10.1093/brain/awx164.

  2. Rappel, P.; Marmor, O.; Bick, A.S.; Arkadir, D.; Linetsky, E.; Castrioto, A.; Tamir, I.; Freedman, S.A.; Mevorach, T.; Gilad, M.; et al. Subthalamic theta activity: a novel human subcortical biomarker for obsessive compulsive disorder. Translational Psychiatry 2018, 8, 118, doi:10.1038/s41398-018-0165-z.

We genuinely appreciate your feedback, as it has enriched our manuscript and made it more comprehensive. We hope that the revisions adequately address your concerns.

Reviewer 4 Report

Thanks for your comment

Author Response

Thank you for recognizing the improvements in our manuscript. We appreciate your positive feedback and are pleased that the added EEG data enhanced the paper's quality.

Best regards,

Ilju, Lee